# Ontology-Based Map Data Quality Assurance

Haonan Qiu[1,2], Adel Ayara[1], and Birte Glimm[2]

[1] BMW Car IT GmbH, Ulm, Germany,
{haonan.qiu,adel.ayara}@bmw.de
[2] Institute of Artificial Intelligence, University of Ulm, Germany,
birte.glimm@uni-ulm.de

**Abstract.** A lane-level, high-definition (HD) digital map is needed for autonomous cars to provide safety and security to the passengers. However, it continues to prove very difficult to produce error-free maps. To avoid the deactivation of autonomous driving (AD) mode caused by map errors, ensuring map data quality is a crucial task. We propose an ontology-based workflow for HD map data quality assurance, including semantic enrichment, violation detection, and violation handling. Evaluations show that our approach can successfully check the quality of map data and suggests that violation handling is even feasible on-the-fly in the car (on-board), avoiding the autonomous driving mode's deactivation.

**Keywords:** Autonomous driving · Digital maps · Ontologies · Rules

## 1 Introduction

Autonomous cars act in a highly dynamic environment and consistently have to provide safety and security to passengers. A detailed, high-definition (HD) map is needed for a car to understand its surroundings, which provides *lane-level* information to support vehicle perception and highly precise localisation [3]. The creation of a road map involves a series of decisions on how features of the road are to be represented concerning the map scale, level of generalization, projection, datum, and coordinate system. Every step of map creation may introduce an error in one of the map features and Figure 1 shows a road gap that has been found in a commercially available HD map, which caused a degradation of the autonomous driving (AD) mode and a driver take-over request.

Usually, a take-over request is conducted for safety reasons when the AD system is approaching its limits due to, for example, weather conditions. In general, a take-over request is a complex and risky process and should be avoided as much as possible and, in case of map errors, the request is not even related to system limits. Therefore, the goal of our work is to use ontologies and reasoning to find and fix map errors to extend the AD function's availability. Ensuring (general) data quality with an ontology-based approach has been well-studied recently [7, 12, 6]. Yilmaz et al. [28, 27] have even demonstrated the feasibility of using ontological methods for spatial data quality evaluation. The latter work does, however, not consider map-specific concepts, e.g., lanes and the resulting

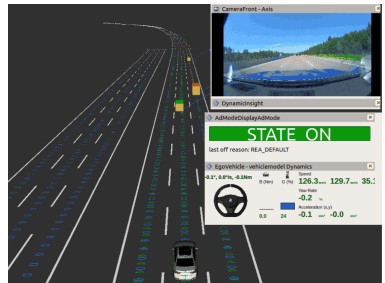 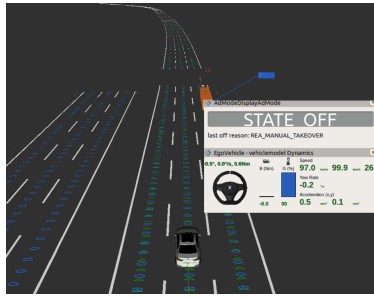

Fig. 1: Snap shots of a normal driving scenario without map errors and active AD mode (left-hand side) and an error scenario (right-hand side) with deactivated AD mode and a driver take-over request due to a gap in the road model

challenges, and neither are the challenges and possibilities of *handling* violations considered. We address these challenges and present an ontology-based approach for ensuring map data quality. The main contributions are:

- We present a workflow for ensuring map data quality based on OWL 2 RL ontologies [13] and Datalog rules [1].
- We present the develop Map Quality Violation Ontology (MQVO) and a set of constraint rules for violation detection.
- We demonstrate violation handling strategies using violation tolerance and resolution.
- We evaluate the performance of violation detection and the correctness of violation resolution using RDFox [21] and realistic map data.

The rest of this paper is structured as follows: Section 2 introduces related work, followed by some preliminaries in Section 3. Section 4 describes the workflow consisting of semantic enrichment, violation detection, and violation handling. In Section 5, we describe the experimental setup and results, and we conclude in Section 6. Additional explanations, rules, available resources and evaluation discussion can be found in an accompanying technical report [22].

## 2   Related Work

Spatial data quality can be assessed with ontology-based approaches. Mostafavi et al. [18] propose an ontology-based approach for quality assessment of spatial databases. The ontology is encoded in Prolog, and queries are used to determine the existence of inconsistencies. Wang et al. [25] investigate the feasibility of applying rule-based spatial data quality checks over mobile data using the Semantic Web Rule Language (SWRL). The authors show that the system has the capability to warn the data collector if there is any inconsistent data gathered in the field. Yilmaz et al. [28] created an ontology associated with spatial concepts from the Open Geospatial Consortium and rules implemented as GeoSPARQL

queries for detecting inconsistencies. Yilmaz et al. also developed the Spatial Data Quality Ontology together with SWRL rules for performing quality assessment [27]. Huang et al. [14] investigate the feasibility of combining ontologies and semantic constraints modelled in the Shapes Constraint Language (SHACL) for ensuring the semantic correctness of geospatial data from different levels of detail. A number of RDF stores also support geospatial queries and integrity constraints, e.g., Stardog,[3] Virtuoso,[4] and GraphDB.[5]

The existing ontology-based approaches, however, focus on general spatial data. Map-related concepts and relationships, such as the relationships among coordinate points, lanes, and roads, are not studied. While SHACL is designed for RDF validation, by checking nodes w.r.t. class axioms or paths w.r.t. property axioms, it cannot describe complex (spatial) relationship constraints, which is crucial for map data. Although SHACL provides validation reports, it does not provide a mechanism (e.g., vocabulary) for fixing errors, while we aim at supporting violation detection and handling in a closed loop.

## 3    Preliminaries

In this section, we present relevant background for map data, the Resource Description Framework, and rules.

### 3.1    Map Data

Our work is focused on the Navigation Data Standard (NDS) [20]. Map data is partitioned in to adjacent *tiles*. They form approximately rectangular territorial sections. The magnification *level* determines the edge length of a tile. *Nodes* within a map tile represent a point location on the surface of the Earth by a pair of longitude (y-coordinate) and latitude (x-coordinate) coordinates. *Links* represent a stretch of road between two nodes and are represented by a line segment (corresponding to a straight section of the road) or a curve having a shape that is generally described by intermediate points called *shape points* along the link. Shape points are represented by x-y coordinates as nodes, but shape points do not serve the purpose of connecting links, as do nodes. Link and road are synonyms and road has the same meaning as in everyday language use. Links have attributes such as travel direction and types, such as highway. The ordering of the shape points is with respect to the travel direction. The geometry of *Lanes* is described by shape points too. Lanes are connected via *lane connectors*. Each lane is described by two *lane boundaries* with lane marking types (solid/dashed, single/double, etc.). Finally, lanes are organized into *lane groups* with link references. We refer the interested reader to the literature for further details about HD maps [15, 11].

---

[3] https://www.stardog.com/
[4] https://virtuoso.openlinksw.com/
[5] https://graphdb.ontotext.com/

### 3.2   RDF Graphs

Resource Description Framework (RDF) is a W3C standardised model for data interchange in applications on the Web, where a subject ($s$) and a object ($o$) are related with an explicit predicate ($p$). These simple *s-p-o* statements can be seen as a directed, labelled (knowledge) graph. We formally introduce RDF graphs as follows:

**Definition 1 (RDF Graph [5]).** *Let $I$, $L$, and $B$ be pairwise disjoint infinite sets of IRIs, literals, and blank nodes, respectively. A tuple $(s, p, o) \in I \cup B \times I \times (I \cup L \cup B)$ is called an RDF* triple, *where $s$ is the* subject, *$p$ is the* predicate, *and $o$ is the* object. *An RDF graph $G$ is finite set of RDF triples and induces a set of* vertices $V = \{s \mid (s, p, o) \in G\} \cup \{o \mid (s, p, o) \in G\}$.

On top of RDF, we use the RL (rule language) profile of the Web Ontology Language (OWL) [13] and custom Datalog rules [1] (RDFox syntactic variant) to model complex knowledge and to infer, in particular, spatial relationships.

### 3.3   Rules

For defining such *Datalog rules*, we fix countable, disjoint sets of *constants* and *variables*. A *term* is a constant or a variable. An *atom* has the form $P(t_1, \ldots, t_k)$, where $P$ is a $k$-ary predicate and each $t_i$, $1 \leq i \leq k$, is a term. We focus on unary and binary atoms only (i.e., $1 \leq k \leq 2$), which correspond to classes and properties of the ontology, respectively. An atom is *ground* if it does not contain variables. A *fact* is a ground atom and a *dataset* is a finite set of facts, e.g., as defined in an ontology. A Datalog *rule* is a logical implication of the form $H_1, \ldots, H_j \leftarrow B_1, \ldots, B_k$, where each $H_i$, $1 \leq i \leq j$, is a *head* atom, and each $B_\ell$, $1 \leq \ell \leq k$, is a *body* atom. A Datalog *program* is a finite set of rules.

A *negative* body atom has the form, `NOT EXISTS` $v_1, \ldots, v_j$ `IN` $B$, where each $v_i$, $1 \leq i \leq j$, is a variable and $B$ is an atom. A rule $r$ is *safe* if variables that appear in the head or in a negative body atom also appear in a positive body atom. A safe Datalog rule can be extended with *stratified negation* by extending the rule to have negative body atoms, where there is no cyclic dependency between any predicate and a negated predicate.

An *aggregate* is a function that takes a multiset of values as input and returns a single value as output. An aggregate atom has the form `Aggregate`($B_1, \ldots, B_k$ `ON` $x_1, \ldots, x_j$ `BIND` $f_1(e_1)$ `AS` $r_1 \ldots$ `BIND` $f_n(e_n)$ `AS` $r_n$), where each $B_i$, $1 \leq i \leq k$, is an atom, each $x_u$, $1 \leq u \leq j$, is a variable that appears in $B_i$, each $f_v$, $1 \leq v \leq n$, is an aggregate function, each $e_w$, $1 \leq w \leq n$, is an expression containing variables from $B_i$, and each $r_z$, $1 \leq z \leq n$, is a constant for a variable that does not appear in $B_i$.

## 4   Ensuring Map Data Quality

In this section, we present the workflow of ensuring map data quality consisting of: (i) semantic enrichment, (ii) violation detection, and (iii) violation handling (see Figure 2). We next describe these steps in more detail.

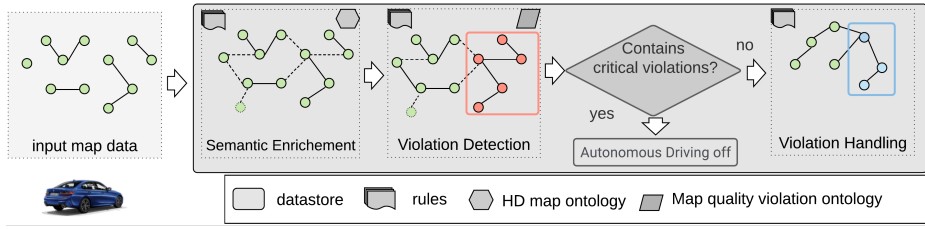

Fig. 2: Workflow diagram of ensuring map data quality

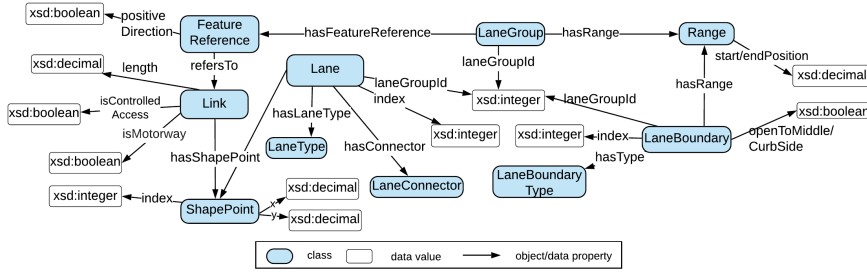

Fig. 3: An HD map ontology based on NDS (partial rendering)

### 4.1  Semantic Enrichment

We adopt the concept of semantic enrichment [10] and use a set of rules for inferring spatial semantics, e.g., start/end points and direct lane successors. This allows us to express complex spatial relationships, which are the basis for the subsequent violation detection and handling process. We modelled an HD map ontology based on the NDS specification for describing the map entities as shown in Figure 3. The rules can be categorised into: (1) primitive rules, (2) bounding rules, (3) coordinate distance rules, and (4) topological rules:

*(1) Primitive rules* enrich instances with one-step inferences regarding relationships and attributes and their results serve as input for all other rules. For a concrete example consider:

hasLane$(x, y) \leftarrow$ LaneGroup$(x)$, laneGroupId$(x, i)$, Lane$(y)$, laneGroupId$(y, i)$.

*(2) Bounding rules* infer the boundaries of an area or the range of a lane or road, such as a start/end shape point of a lane or the left or right-most lane. As a concrete example, consider:

StartShapePoint$(z) \leftarrow$ Lane$(l)$, AGGREGATE(hasShapePoint$(l, p)$, index$(p, idx)$ ON $l$
BIND MIN$(idx)$ AS $m$), hasShapePoint$(l, z)$, index$(z, m)$.

AGGREGATE takes the matches for hasShapePoint$(l, p)$ and index$(p, idx)$ as input and groups them based on the lane $l$. The aggregation function MIN then selects the minimal index per group and assigns this value to $m$ using BIND.

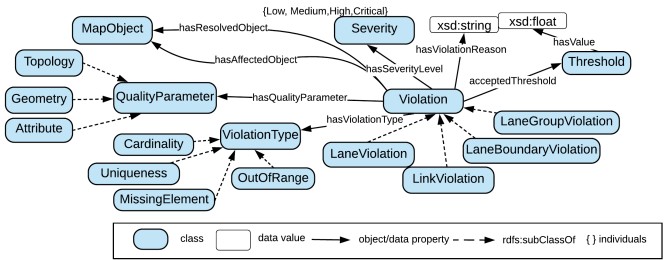

Fig. 4: Classes and properties in MQVO

Since there is no suitable aggregate function for selecting the point for the index $m$, it remains to get the point $z$ that has the index $m$.

*(3) Coordinate distance rules* indicate the distance between two points using coordinates. An auxiliary concept (CoordinateDistance) represents the ternary relation that connects the source point to the target point via two object properties hasSource and hasTarget and the calculated distance value via the data property distance:

$$\mathsf{CoordinateDistance}(d),\ \mathsf{hasSource}(d,s),\mathsf{hasTarget}(d,t),\ \mathsf{distance}(d,z) \leftarrow$$
$$\mathsf{ShapePoint}(s),\ \mathsf{x}(s,x_s),\ \mathsf{y}(s,y_s),\ \mathsf{ShapePoint}(t),\ \mathsf{x}(t,x_t),\ \mathsf{y}(t,y_t),$$
$$\mathtt{BIND}(\mathtt{sqrt}((x_s-x_t)^2+(y_s-y_t)^2)\ \mathtt{AS}\ z),\mathtt{BIND}(\mathtt{SKOLEM}("d",\ s,\ t)\ \mathtt{AS}\ d).$$

The `SKOLEM` function allows for dynamically generating "fresh" IRIs [5] based on the string `"d"` and the variable bindings for $s$ and $t$.

*(4) Topology rules* refer to topological relations, more specifically, lateral (left-/right) and longitudinal (predecessor/successor) relations. Connectivity can naturally be expressed using recursive rules. The base case (one-step connectivity) is usually inferred based on a pre-defined connectivity reference. For example, the `hasDirectNext` relation over lanes is defined based on source and destination connectors, while for links, it is defined by comparing the coordinates of start and end points of links.

### 4.2   Violation Detection

We developed the Map Quality Violation Ontology (MQVO) and a set of spatial constraint rules to detect violations after the enrichment process. The MQVO describes the type of violation, the affected objects, the severity level, etc. and provides information to guide the subsequent violation handling process. The spatial constraint rules are classified into (1) topology, (2) geometry and (3) attribute checking rules. We first describe MQVO, then we introduce the three types of constraint checking rules.

*Map Quality Violation Ontology* We developed MQVO to describe map data errors since, to the best of our knowledge, there are no ontologies for the specific purpose of describing map data violations. MQVO supports error detection by

Table 1: Constraint axioms as Datalog Constraint Atoms (DCA). We use $\mathsf{C}$ for classes, $\mathsf{op}$ for object, $\mathsf{dp}$ for data, and $\mathsf{p}$ for object or data properties.

| OWL Axiom | Datalog Constraint Atom (DCA) |
|---|---|
| Existential Quantification | $\mathsf{C}(x)$, $\mathtt{NOT\ EXISTS}\ y\ \mathtt{IN}\ (\mathsf{C}(y),\ \mathsf{p}(x,\ y))$ |
| Individual Value Restriction | $\mathsf{C}(x)$, $\mathtt{NOT}\ \mathsf{op}(x,\ \mathit{individual})$ |
| Literal Value Restriction | $\mathsf{C}(x)$, $\mathtt{NOT}\ \mathsf{dp}(x,\ \mathit{literal})$ |
| HasKey | $\mathsf{C}(x)$,  $\mathsf{dp}(x,z)$, $\mathsf{C}(y)$, $\mathsf{dp}(y,z)$, $\mathtt{FILTER}(x \neq y)$ |
| Min $(<)$/Max $(>)$ | $\mathsf{C}(x)$, $\mathtt{AGGREGATE}(\mathsf{p}(x,v)\ \mathtt{ON}\ x\ \mathtt{BIND}\ \mathtt{count}(v)\ \mathtt{AS}\ n)$, |
| Cardinality Restriction | $\mathtt{FILTER}(n \bowtie max), \bowtie \in \{>,<\}$ |

defining properties that can identify map objects (e.g., links/lanes) in which a violation is detected. It also provides context information for a violation to guide the repair process. Figure 4 shows the main concept `Violation` and the related properties and classes. Each violation is associated with a `ViolationType`, `QualityParameter`, `Severity`, affected `MapObject`, and the reason. `Severity` is described by one of the individuals `Low`, `Medium`, `High` and `Critical`. If a violation is repaired, then the involved map objects are linked to it via the `hasResolvedObject` object property. A violation can have an accepted `Threshold`, such as the threshold of the distance between two points.

*Constraint Rules* are classified into (1) topology, (2) geometry, and (3) attribute checking rules based on the map error types. Before describing the details of each rule type, we first introduce *Violation Recording Rule Templates* (VRRTs), which provide patterns for modelling constraint violation detection rules. Table 1 shows OWL axioms used to capture the map data quality requirements together with their corresponding *Datalog Constraint Atoms* (DCA). Constraint violations are recorded using freshly generated instances of `Violation` as shown in the following rule template:

$$\mathsf{Violation}(v),\ \mathsf{hasAffectedObject}(v,x),\ \mathsf{hasReason}(v,\texttt{"r"}) \leftarrow$$
$$<\!\text{DCA}\!>,\ \mathtt{BIND}(\mathtt{SKOLEM}(\texttt{"}d\texttt{"},\ x)\ \mathtt{AS}\ v).$$

A concrete example of minimum cardinality constraint of lane shape points using the above template is as follows:

$$\mathsf{Violation}(v),\ \mathsf{hasAffectedObject}(v,l),\ \mathsf{hasReason}(v,\texttt{"MinCardinalityError"}) \leftarrow$$
$$\mathsf{Lane}(l),\ \mathtt{AGGREGATE}(\mathsf{hasPoint}(l,p)\ \mathtt{ON}\ l\ \mathtt{BIND}\ \mathtt{count}(p)\ \mathtt{AS}\ n),$$
$$\mathtt{FILTER}(n < 2),\ \mathtt{BIND}(\mathtt{SKOLEM}(\texttt{"}d\texttt{"},\ l)\ \mathtt{AS}\ v).$$

*(1) Topology Checking Rules* are designed for checking the spatial relationships of map objects. A comprehensive formal categorisation of binary topological relations between regions, lines, and points has been developed by Egenhofer and Herring [9]. In this paper, we model *full coverage* constraints, checking if a set of other map objects fully covers a given map object. For example, a link should be fully covered by a set of lane groups. To describe such constraints, we first introduce some basic notations.

---

**Algorithm 1:** Check full coverage

    **input** : $\overline{pq}$: a base line, $L = \{\overline{u_1 v_1}, \ldots, \overline{u_n v_n}\}$: a set of line segments
    **output:** $L_g, L_o$: sets of line segments causing gaps and overlappings, resp.

**1** $L_g = L_o = \emptyset$;

**2** $\overline{u_s v_s} = \text{GETSTARTSEGMENT}(L)$ ;                          `// apply bounding rules`

**3** $\overline{u_e v_e} = \text{GETENDSEGMENT}(L)$ ;                           `// apply bounding rules`

**4** **if** $u_s > p$ **then**

**5**     $L_g = L_g \cup \{\overline{u_s v_e}\}$ ;                                `// gap at the start`

**6** **if** $v_e < q$ **then**

**7**     $L_g = L_g \cup \{\overline{u_e v_e}\}$ ;                                `// gap at the end`

**8** **for** $i = 1 \ldots n - 1$ **do**

**9**     $\overline{uv} = \text{GETDIRECTNEXT}(\overline{u_i v_i})$ ;                  `// apply topology rules`

**10**     **if** $v_i < u$ **then**

**11**        $L_g = L_g \cup \{\overline{u_i v_i}, \overline{uv}\}$ ;                   `// gap in the middle`

**12**     **else if** $v_i > u$ **then**

**13**        $L_o = L_o \cup \{\overline{u_i v_i}, \overline{uv}\}$ ;                       `// overlapping`

---

**Definition 2.** *A* line (segment) $\overline{pq}$ *is defined by its* start point $p$ *and its* end point $q$, *where* $p \neq q$. *A (base) line* $\overline{pq}$ *is* fully covered *by a sequence of lines* $\overline{u_1 v_1} \ldots \overline{u_n v_n}$ *if* $p = u_1$, $q = v_n$, *and* $v_i = u_{i+1}$ *for each* $1 \leq i < n$, *where* $u_i \neq v_i$. *We say that there is a* gap at the start *if* $p < u_1$, *a* gap at the end *if* $v_n < q$, *a* gap in the middle *if* $v_i < u_{i+1}$ *for some* $1 \leq i < n$, *and there is an* overlapping *if* $v_i > u_{i+1}$ *for some* $1 \leq i < n$.

Algorithm 1 presents the pseudo-code of full coverage checking. Given a base line $\overline{pq}$ and a sequence of line segments $L$ without self-loops, the algorithm first identifies the start and end segment in $L$ (lines 2–3) using bounding rules. The algorithm checks if there is a gap at the start or end w.r.t. the base line (lines 4–7). At last, it iterates through the given line segments and, for each segment, it gets the direct next line segment (line 9) through topology rules, checks if there is any gap in the middle (lines 8–10) or any overlapping (lines 11–12).

*(2) Geometry Checking Rules* are designed to check the geometric representation of links (lanes). The link (lane) model uses an ordered sequence of shape points describing the geometry of a polyline that represents a link (lane). We further subdivide geometry checking rules into *cardinality* and *geometric accuracy* checking rules. Cardinality checking rules use minimum or maximum cardinality restrictions in VRRTs. Geometric accuracy is checked via coordinate proximity using distance thresholds [8] to account for different levels of accuracy and precision [29] in the collected map data. The following rule, where we abbreviate hasAffectedObject as hao, illustrates the case of checking a radius distance threshold of geometric points in two connected lanes.

Violation($v$), hao($v, u_p$), hao($v, v_q$), hasReason($v$, "GeometryError") $\leftarrow$
       Lane($p$), Lane($q$), hasDirectNext($p, q$), endPoint($p, u_p$), startPoint($q, v_q$),

CoordinateDistance$(c)$, hasSource$(c, q)$, hasTarget$(c, p)$, distance$(c, d)$,
Threshold$(t)$, hasValue$(t, v_t)$, FILTER$(d > v_t)$,
BIND(SKOLEM("$d$", $u_p$, $v_q$) AS $v$).

*(3) Attribute Accuracy Checking Rules* are used to check if the recorded attributes of map data representing real-world entities are correct and consistent. The attributes could be feature classifications, text information for feature names, or descriptions, and they ought to be consistent with each other. For example, if a road is classified as a motorway, it should also have a controlled-access designed for high-speed vehicular traffic. Controlled-access is modelled as a data property with a Boolean value. Hence, the corresponding violation detection rule can be modelled using a literal value restriction in the VRRT.

### 4.3 Violation Handling

Violations are handled based on the severity level. If a critical violation is detected during the map pre-loading phase, the autonomous driving mode is switched off, and control is handed over to the driver in the corresponding region. For non-critical violations, we rely on *violation tolerance* and *violation resolution* strategies considering the spatial relations. Violation tolerance is feasible because errors in the low-level (raw) data do not necessarily affect the decision taken at the knowledge (human-perceivable) level in intelligent systems [26]. In cases where the violations cannot be tolerated, spatial knowledge, e.g., topological relations, can be used to resolve violations [17, 2]. These strategies allow us to support autonomous driving applications, even in the presence of low-level data errors.

We apply graph aggregation [16] for violation tolerance and decomposition [4] for violation resolution to achieve knowledge level consistency. Essentially, these strategies take advantage of graph structure similarity, which is captured by the notion of *isomorphisms* :

**Definition 3 (RDF Graph Isomorphism [5]).** *Let $G_1$ and $G_2$ be RDF graphs with $V_1, V_2$ the induced vertices of $G_1$ and $G_2$, respectively. We say that $G_1$ and $G_2$ are* isomorphic*, if there is a bijection $\mu : V_1 \to V_2$ such that $\mu(b) \in B$ for each $b \in V_1 \cap B$, $\mu(\ell) \in L$ for each $\ell \in V_1 \cap L$, $\mu(v) \in I$ for each $v \in V_1 \cap I$, and, for each triple $(s, p, o) \in G_1$, $(\mu(s), p, \mu(o)) \in G_2$. We call such $\mu$ an* isomorphism *between $G_1$ and $G_2$.*

Based on isomorphism, we introduce graph aggregation and its use for *violation tolerance*. Apart from its use in violation tolerance, graph aggregation is helpful in itself to obtain a higher-level view of the map data, with a focus on the details that are important for autonomous driving.

**Definition 4 (RDF Graph Aggregation).** *Let $G_1$, $G_2$, and $G$ be RDF graphs with vertices $V_1$, $V_2$, and $V$, respectively, such that $G_1$ is isomorphic to $G_2$ witness by the isomorphism $\mu$. A (partial) function $\alpha : V_1 \cup V_2 \to V$ is an* abstraction function *w.r.t. $G_1$, $G_2$, and $G$ if, for each $v \in V$, there are nodes $v_1 \in V_1$ and $v_2 \in V_2$ such that $\mu(v_1) = v_2$ and $\alpha(v_1) = \alpha(v_2) = v$. If an abstraction function w.r.t. $G_1$, $G_2$, and $G$ exists, we call $G$ an* aggregation graph *of $G_1$ and $G_2$.*

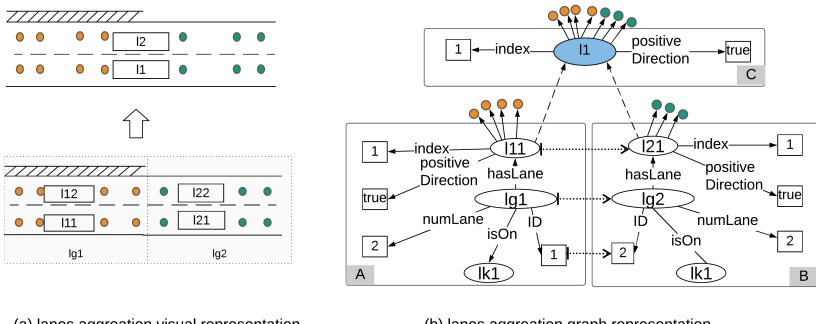

(a) lanes aggregation visual representation.          (b) lanes aggregation graph representation

Fig. 5: A violation-free example of RDF graph aggregation over lanes

We generalise the notion of an abstraction graph to a set of pairwise isomorphic graphs $G_1, \ldots, G_n$ in a natural way.

Figure 5 shows an example where we apply graph aggregation over lanes of two lane groups with ID 1 and 2. We abbreviate lane as l, laneGroup as lg, and link as lk, e.g., l11 stands for lane11. Subfigure (a) shows a map visualisation, while (b) shows the corresponding graph representation, and the aggregation is shown in the upper part. Note that the mapping with a dotted line shows the isomorphism between graph A and graph B (we omit the mapping for identical values such as a mapping from true in graph A to true in graph B). The dashed lines show the abstraction function, where we again omit identical value mappings. The abstraction function only maps the lanes (l11 in graph A and l21 in graph B) as well as the lane index and direction attribute. The lane aggregation aligns with the human perception of l11 and l21 as one continuous lane.

*(1) Violation Tolerance.* Figure 6 shows an example with a violation, which consists of a duplicate lane group ID. More precisely, lg1 and lg2 both have ID 1 in the map data. As a result of this, the map data is parsed as containing just one lane group (with ID 1), which also causes l11 and l21 to be considered equal as they both have ID 1 and belong to the lane group with ID 1. Hence, we get identical RDF graphs for l11 (graph A) and l21 (graph B), which is a special case of RDF graph isomorphism. Applying the abstraction function (as in Figure 6, dashed line) results, however, in the same (correct) aggregation graph (graph C) as for the violation-free scenario shown in Figure 5. Hence, the RDF graph aggregation can tolerate some data errors.

*(2) Violation Resolution.* We illustrate how violations can be resolved (in particular, lane ambiguity) using graph decomposition.

**Definition 5 (RDF Graph Decomposition).** *An* RDF decomposition *of an RDF graph $G$ is a collection of edge-disjoint, isomorphic subgraphs $G_1, \ldots, G_n$ of $G$ such that every edge of $G$ belongs to exactly one $G_i$, $1 \leq i \leq n$. We denote such a decomposition of $G$ as $\hat{G} = \{G_1, G_2, \ldots, G_n\}$.*

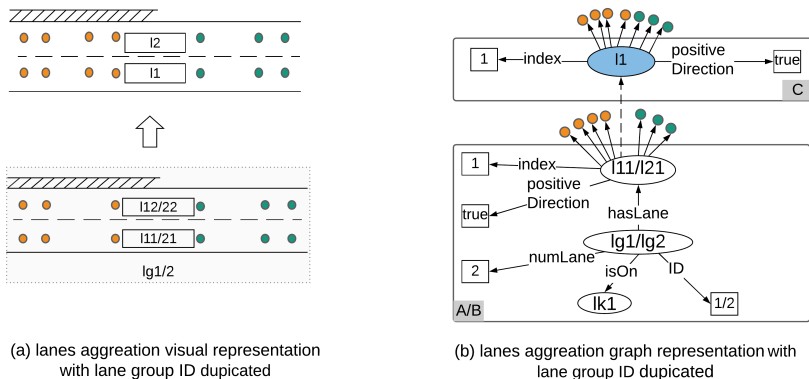

(a) lanes aggregation visual representation with lane group ID dupicated

(b) lanes aggregation graph representation with lane group ID dupicated

Fig. 6: An example of lane aggregation with lane group ID uniqueness violation

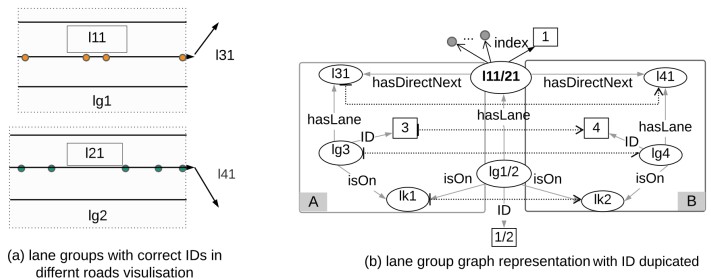

(a) lane groups with correct IDs in differnt roads visulisation

(b) lane group graph representation with ID dupicated

Fig. 7: A example of lane ambiguity caused by lane group ID duplication.

Figure 7 shows an example of lane ambiguity also caused by a lane group ID duplication. Subfigure (a) shows a normal map visualisation of lg1 and lg2 located in separate roads. Subfigure (b) shows the graph representation resulting from a duplicate ID of lg1 and lg2 which causes l11 and l21 to merge into one lane instance having both lanes' spatial relationships, such as associated points, links and successor lanes. Based on the graph structure of the ambiguous graph, there exists a mapping between subgraph A and B, which indicates the application of RDF decomposition. Hence, we apply RDF graph decomposition to fix the topology and distance measurements to restore geometry. Figure 8 shows the concrete steps: (1) violation detection, (2) topology correction, and (3) assignment of geometric points.

In Step 1, a topology violation is detected if a lane group is associated with two disconnected links. This is modelled by checking the existence of a connection between links associated to a lane group using an existential qualification in a VRRT, and an instance of LaneViolation is generated having topology (an instance of class Topology) as its QualityParameter.

In Step 2, the topology correction is achieved via graph decomposition and relationship establishment. The original graph of l11/21 can be decomposed

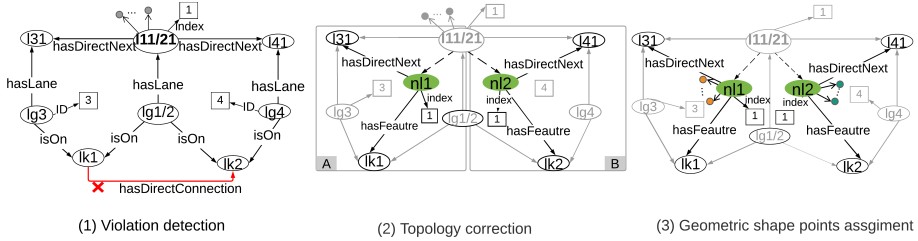

(1) Violation detection        (2) Topology correction        (3) Geometric shape points assgiment

Fig. 8: Lane ambiguity violation resolution steps

into isomorphic subgraphs A and B. Two new lane instances (nl1 and nl2) are generated with the correct topological relationships.

$$\text{NewLane}(l), \text{hasFeature}(n, f), \text{hasDirectNext}(l, n), \text{hasOriginalLane}(l, m) \leftarrow$$
$$\text{LaneViolation}(v), \text{hasQualityParameter}(v, \text{topology}), \text{hao}(v, m),$$
$$\text{hasLane}(lg_1, m), \text{isOn}(lg_1, f), \text{LaneGrp}(lg_2), \text{hasLane}(lg_2, n),$$
$$\text{isOn}(lg_2, f), \text{hasDirectNext}(m, n), \text{index}(m, i),$$
$$\text{BIND}(\text{SKOLEM}("d", f, i) \text{ AS } l)$$

In Step 3, geometric shape point assignment is achieved via a point grouping strategy which compares the distance from each shape point of the lane to the first and last shape point of the lane associated links. The shape points are then grouped if the difference between the calculated distance and the links' length is within a threshold, e.g., $10\,m$.

$\text{hasPossibleLanePoint}(f, p) \leftarrow$
$$\text{LaneViolation}(v), \text{hasQualityParameter}(v, \text{topology}), \text{hao}(v, l), \text{hasShapePoint}(l, p),$$
$$\text{hasFeature}(l, f), \text{length}(f, n), \text{hasFirstShapePoint}(f, u), \text{hasLastShapePoint}(f, v),$$
$$\text{CoordinateDistance}(d_1), \text{hasSource}(d_1, p), \text{hasTarget}(d_1, u), \text{distance}(d_1, t_1),$$
$$\text{CoordinateDistance}(d_2), \text{hasSource}(d_2, p), \text{hasTarget}(d_2, v), \text{distance}(d_2, t_2),$$
$$\text{FILTER}(\text{ABS}((t_1 + t_2) - n) < 10).$$

While assigning the point groups to correct new lanes, geometric point grouping is verified by comparing the number of points in each group to the total number of points of the original lane to prevent wrong point group assignments.

$\text{hasShapePoint}(n, p) \leftarrow$
$$\text{NewLane}(n), \text{hasOriginalLane}(n, l), \text{numPoints}(l, u), \text{hasFeature}(n, f),$$
$$\text{hasPossibleLanePoint}(f, p), \text{numPossibleLanePoints}(f, m), \text{FILTER}(m < u).$$

## 5   Evaluation

Tile-based map data is stored as Binary Large Object (BLOB) in an NDS map database. We use SQLite Python APIs to extract map data and construct RDF triples based on the HD map ontology (see Figure 3). We have implemented the proposed workflow of ensuring map data quality into an application called *SmartMapApp* using RDFox 4.0.0 as reasoner. The evaluation was performed on a 64-bit Ubuntu virtual machine with 8 Intel(R) Core(TM) i7-8550U CPU

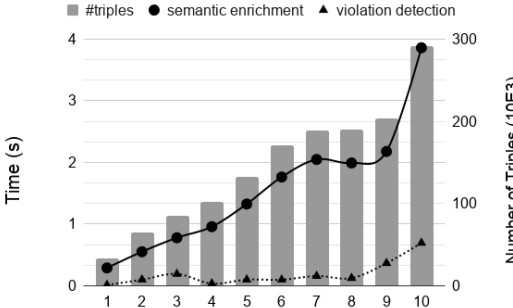

Fig. 9: Performance of semantic enrichment and violation detection over real map data; the left-hand side scale shows the execution time for semantic enrichment (dots on a solid line) and violation detection (diamonds on a dotted line) in seconds; the number of input triples is shown in form of bars using the scale on the right-hand side

Table 2: Number of rules used for semantic enrichment and violation detection

| Semantic Enrichment Rules | | Violation Detection Rules | |
|---|---|---|---|
| Primitive Rules | 15 | Topology Checking Rules | 10 |
| Bounding Rules | 14 | Geometry Checking Rules | 14 |
| Coordinate Distance Rules | 3 | Attribute Accuracy Checking Rules | 13 |
| Topology Rules | 10 | Total | 37 |
| Total | 42 | | |

@ 1.80GHz running at 33MHz with 15 GB memory. We first show the performance of semantic enrichment and violation detection and then we evaluate the correctness of violation handling.

## 5.1  Violation Detection

We used 10 adjacent real map tiles along Federal Motorway 92 (Bundesautobahn 92) in Germany for violation detection evaluation, and record the computation time after doing a warm-up run by executing the tasks 3 times sequentially . Semantic enrichment is performed via 42 rules and violation detection consists of 37 rules (see Table 2). The result of the two phases is summarised in Figure 9. The computation time for both phases increases with respect to the data size. The average number of input triples is $146,182$, the average execution time of semantic enrichment is $1,584\,\mathrm{ms}$, and the average execution time of violation detection is $197\,\mathrm{ms}$.

Table 3: The violation tolerance over a lane using graph aggregation, both ground truth and dirty data are aggregated results

|  | lane | length (m) | #points | successor | link |
|---|---|---|---|---|---|
| ground truth | BE9D6 | 2034 | 54 | 563E | 02 |
| dirty data | BE9D6 | 2034 | 54 | 563E | 02 |

Table 4: The violation resolution of a lane using graph decomposition

|  | lane | length (m) | #points | successor | link |
|---|---|---|---|---|---|
| ground truth | 1116_0 | 358 | 14 | 1117_0 | 199 |
| dirty data | 1116_0 | 214, 358 | 5, 14 | 1117_0, 1188_0 | 199, 197 |
| resolved violation | 1116_0 | 358 | 14 | 1117_0 | 199 |

### 5.2   Violation Handling

We consider the use case of lane group ID uniqueness violations to evaluate violation handling strategies. We show the result of violation tolerance over the error on a high way described in Figure 1 (see Section 1), and violation resolution over an error on separated roads. At last, we discuss the evaluation results.

*Violation Tolerance.* The error in Figure 1 occurred in the map data containing a highway in Germany. Part of this highway is represented as five continuous lane groups with the same number of lanes. Two of the lane groups have the same ID, which caused the degradation of the autonomous driving mode. We applied graph aggregation over both the ground truth and dirty data. The lane aggregation results agree on both inputs, which shows that the lane group ID issue can be solved (see Table 3).

*Violation Resolution.* We evaluated the resolution strategy over two lane groups containing only one lane allocated to different links. The ID duplication caused the lanes in these two lane groups to be merged into one lane. Table 4 shows the result of applying graph decomposition over the dirty data. Row 2 (dirty data) shows that the lane is ambiguous as it has the length, number of shape points, successors, and related links of both lanes. Row 3 shows that the graph decomposition can resolve the error and all lane properties are correctly recovered.

Overall, the results demonstrate that our approach can improve map data quality, resulting in a better error-tolerance of AD systems. On the one hand, the performance of the violation detection allows the deployment of the proposed solution in the back-end (cloud side) to check the map data before sending it to the car or on-board (embedded side, in the car) in case of loss of connectivity with the back-end. On the other hand, the evaluation of the violation handling strategies has shown that we could avoid the deactivation of the AD mode by detecting the error and correcting the map data in both cases of the lane group error. The cost of reasoning generally depends not only on the number of rules but also on the complexity of the combination of certain rules and the input

data. For details of how RDFox performs reasoning, we refer interested readers to the description of the materialization algorithm in RDFox [19].

## 6    Conclusion and Future Work

In this paper, we present an ontology-based approach for ensuring map data quality. We propose a workflow considering semantic enrichment, violation detection and violation handling. Semantic enrichment is achieved via a set of rules combined with an HD map ontology and the results provide the needed spatial knowledge for violation detection and handling. Violation detection is modelled via the novel Map Quality Violation Ontology and suitable constraint violation rules. At last, we show novel violation handling strategies over non-critical violations using graph aggregation and graph decomposition. We evaluate the performance of violation detection and the correctness of violation handling. The results show that our approach can successfully check the quality of map data and suggests that violation handling is even feasible on-the-fly in the car (on-board), avoiding the autonomous driving mode's deactivation. We plan to integrate this approach into the developed knowledge-spatial architecture [23], and test the approach in ROS (Robot Operating System) [24], which requires a re-implementation of the Java-based SmartMapApp in C++.

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
