# OpenReview forum: "Ontology-Based Map Data Quality Assurance"
_eswc-conferences.org/ESWC/2021/Conference/Research_Track — ESWC 2021 Research_

### Official Review · AnonReviewer2 · 2021-01-11
**An application on ontology-based data QA**

**Rating:** 1
**Confidence:** 3
**Impact:** 3
**Design And Technical Quality:** 3

**Review:**

This manuscript presents an interesting application on ontology-based quality assurance for high-definition map data for the application of autonomous driving. The ontology contribution includes semantic enrichment (e.g., using rules for inference), violation detection by a spatial constraint rules (based on RDFox's Datalog rules) and (partial) violation handling. This word be interesting to ESWC participants as a case study of ontology-based data quality.

Strong Points
1) A very interesting application
2) Violations are not only detected but also (partially) solved.
3) Efficiency and scalability of reasoning is carefully considered

Weak Points

1) Data and rules are not open. This would harm the work's impact.

2) More evaluation and data information can be added. The statistics and background of the map data used for evaluation are not clear. What's the computation time for violation handing? Any statistics on the results of violation handling? More examples on violation resolution as in Table 4 would also be helpful. P.S. why the two rows (ground truth and dirty data) are the same in Table 3?

3) More background on HD map data and how they are modeled as RDF triples (ontology) are needed. In Section 3.1, what is the localisation model? What are the original format of map data and how are they represented (transformed into) RDF triples? Meanwhile, some terms used in the rule examples (e.g., the concepts Point and CoordinateDistance) are missing in Fig. 3 (the HD map ontology). The missing of those basic concepts for modeling map data would make it harder to follow the methodology part.



**Anonymity:**

Yes, I would like my review to remain anonymous.

**Reuse And Availability:**

2: Low

**Subreviewer:**

I submitted this review.

---

> ### Author Rebuttal · Authors · 2021-01-29
>
> ## Response to AnonReviewer2
>
> Thank you for your helpful comments, which we will carefully address should the paper be accepted. We address raised questions below:
>
> &quot;_Data and rules are not open. This would harm the work&#39;s impact.&quot;_
>
> We are planning to make the ontology and rules available once the BMW internal review process is completed. The map data is, unfortunately, property of the map data provider (e.g. HERE), but we hope that other users of map data can already benefit from the ontology and rules.
>
> &quot;_More evaluation and data information can be added. The statistics and background of the map data used for evaluation are not clear. What&#39;s the computation time for violation handing? Any statistics on the results of violation handling? More examples on violation resolution as in Table 4 would also be helpful. P.S. why the two rows (ground truth and dirty data) are the same in Table 3?&quot;_
>
> The reason that we did not include the computation statistics of violation handling is that we want to emphasise the ability to fix the errors using rules and context knowledge, whereas the state of the art does not support this. We do have performance statistics for violation tolerance and resolution. Dataset #1 takes 190ms for tolerance and 100ms for resolution; dataset #10 takes 419ms for tolerance and 943ms for resolution. We will consider adding more examples of violation resolution. Table 3 shows the result after aggregation. Hence, both rows have the same result, which is the desired result. The ground truth data shows no lane group ID duplication, while the dirty data does. Having the same aggregation result shows that graph aggregation can tolerate errors in the data such as lane group ID duplication.
>
> &quot;_More background on HD map data and how they are modeled as RDF triples (ontology) are needed. In Section 3.1, what is the localisation model? What are the original format of map data and how are they represented (transformed into) RDF triples?_ _Meanwhile, some terms used in the rule examples (e.g., the concepts Point and CoordinateDistance) are missing in Fig. 3 (the HD map ontology). The missing of those basic concepts for modeling map data would make it harder to follow the methodology part.&quot;_
>
> We will add more background on HD maps and how they are modelled as RDF triples. The localisation model is designed for aiding vehicle localisation using the position of lane boundaries, traffic lights, traffic signs or sensors like LiDAR [1]. The original format of the map data is stored as Binary Large Object (BLOB), and we use SQLite APIs in Python to extract the data and construct RDF triples. The concept Point is the concept ShapePoint in Fig.3. Originally, we used ShapePoint in the rules as well. For the sake of brevity, we renamed ShapePoint to Point in the rules. We will make this consistent in the paper. CoordinateDistance is an auxiliary concept we use to model k-nary relationship for length calculation. However, it is not an original concept from the HD map format. Hence we did not show in the HD map ontology.
>
> References:
>
> [1] Liu, R., Wang, J., Zhang, B.: High definition map for automated driving: Overview and analysis. Journal of Navigation 73(2), 324–341 (2020)

---

### Official Review · AnonReviewer1 · 2021-01-13
**Interesting use case, but with technical issues**

**Rating:** 1
**Confidence:** 3
**Impact:** 4
**Design And Technical Quality:** 3

**Review:**

The paper describes a use of ontologies, datalog rules and RDFox to detect and correct errors in maps user for automated driving.


**Anonymity:**

No, I would like my review to be deanonymized.

**Reuse And Availability:**

2: Low

**Strong Points:**

Interesting and useful use case. Ontology and Datalog rules are interesting and useful.
The decomposition of a complicated problem into Datalog rules makes for good reading.
If not successful here, there is great potential for both the in-use track and the resource track, although the paper
does not fulfill the criteria of these tracks as-is.


**Subreviewer:**

I submitted this review.

**Weak Points:**

Although much effort has been made to explain this complicated domain and to relate it to semantic technology,
the paper still lacks sufficient clarity. There is domain detail that is never made precise.

Especially, sections 3.1 and the first two paragraphs in 4.3 should be much more precise. E.g. what is the relation between a shape point and a link. Does "road" mean the
same as in everyday langauge, or do you actually mean the single-direction half of a normal road? What is the ordering of the shape points in a road exactly? What is a "future map tile"? What does "Data-level violation tolerance is feasible" mean? I would also suggest to only introduce concepts that are necessary for
the presentation, and just make a note that this is a simplification.

Figures 2 and 3 should be "anchored" in the text, and especially figure 2 should be explained better. Figure 3 shows an excerpt of a
map ontology which does not cover the parts used in the examples. This is confusing.

Algorithm 1. I think this should be called "full coverage" not "full converge". This algorithm is not correct, since two disconnected "loops", one containing the start point, and one containing the end point, will pass the alg., but does not satisfy def. 2.

I have not put large weight on this, but I am struggling to see which topic in the call this paper actually fits. There are probably some design patterns here, but they are not clearly identified. The closest topic is probably "Ontologies, schemas, and vocabularies for specific domains (where the emphasis is on the ontology engineering and execution of a methodology)", however, there is no mention of the methodology.

Definition 1: Please state explicitly what the edges are in the RDF Graph.

Definition 4: Why is the abstraction function not restricted to map into the same node types ("Blank", "IRI", Literal) in the same way as the isomorphism is? Why are not the edges part of the abstraction? In the examples this seems like it should be the case, so not sure why.
It is not well motivated why the Severity in MQVO is a string attribute, and not something more structured, e.g. a subclass or key.

---

> ### Author Rebuttal · Authors · 2021-01-29
>
> **Response to AnonReviewer1**
>
> Thank you for your helpful comments, which we will carefully address should the paper be accepted. We address raised questions below:
>
> &quot;_Especially, sections 3.1 and the first two paragraphs in 4.3 should be much more precise. E.g. what is the relation between a shape point and a link. Does &quot;road&quot; mean the same as in everyday language, or do you actually mean the single-direction half of a normal road? What is the ordering of the shape points in a road exactly? What is a &quot;future map tile&quot;?&quot;_
>
> We will carefully analyze which terms are really needed and introduce them in more detail.
>
> Map data is partitioned in to adjacent _tiles_. _Nodes_ within a map tile represent a point location on the surface of the Earth by a pair of longitude (lon) and latitude (lat) coordinates. _Links_ represent a stretch of road (road indeed has the typical meaning also in map data) between two nodes and are represented by a line segment (a straight section of road) or a curve having a shape that is generally described by intermediate points called _shape points_. Shape points are represented by lon-lat coordinates, but shape points do not connect links, as do nodes. Road has the same meaning as in everyday language use. The ordering of the shape points is with respect to the travel direction.For two-way roads, the shape points are ordered based on either one direction.
>
> High definition maps are stored in the cloud, however, so detailed that one cannot load all map data for a country (and beyond) at once within a car. Instead the location of the car is used to determine the tile in which the car is located. This tile and adjacent tiles are then loaded into the system. When the car approaches the borders of a tile, further adjacent tiles (we referred to that as _future tiles_) are loaded.
>
> &quot;_What does &quot;Data-level violation tolerance is feasible&quot; mean? I would also suggest to only introduce concepts that are necessary for the presentation, and just make a note that this is a simplification._&quot;
>
> With &quot;data level violation tolerance is feasible&quot;, we meant that errors in the raw data must not necessarily affect the application that is using the map data. Some errors can be fixed based on context knowledge and reasoning. For example, knowledge about the (spatial) relationships can be used to fix lane group ID errors.
>
> &quot;_Algorithm 1 is not correct, since two disconnected &quot;loops&quot;, one containing the start point, and one containing the end point, will pass the alg., but does not satisfy def. 2.&quot;_
>
> Good point. The self-looping lines are detected using another constraint rule, but we admittedly need to make this clear in the paper. The reason is that self-loops should trigger a &quot;self-looping violation&quot; and not a &quot;full coverage violation&quot; to make the cause of the error more specific.
>
> &quot;_I have not put large weight on this, but I am struggling to see which topic in the call this paper actually fits. There are probably some design patterns here, but they are not clearly identified. The closest topic is probably &quot;Ontologies, schemas, and vocabularies for specific domains (where the emphasis is on the ontology engineering and execution of a methodology)&quot;, however, there is no mention of the methodology.&quot;_
>
> We most see our work in the area of &quot;Ontologies and Reasoning&quot;, which says &quot;this track aims to attract innovative research on ontologies, schemas, vocabularies, the methodologies used to develop them, and logic-based reasoning.&quot; The bullets in the CfP are explicitly not meant as a closed list (&quot;Topics of interest include, but are not limited to&quot;).
>
> &quot;_Definition 4: Why is the abstraction function not restricted to map into the same node types (&quot;Blank&quot;, &quot;IRI&quot;, Literal) in the same way as the isomorphism is? Why are not the edges part of the abstraction? In the examples this seems like it should be the case, so not sure why.&quot;_
>
> There are two reasons for not restricting to nodes types: 1) We do not have blank nodes in our map data, as we would like to know precisely what the node is; 2) there are situations where a literal is mapped to an IRI in order to form relationships.  For example, assume that two lanes (_l11, l21_) are located in a link (_lk1_) and _lk1_  has a triple as _:lk1 :isMotorway &quot;true^^xsd:boolea_n&quot;. After aggregating _l11_ and _l21_ into the lane _l1,_  l1 has the triple _:l1 rdf:type :MotorwayLane_. In this case, the literal _&quot;true^^xsd:boolean&quot;_ is mapped to the IRI _:MotorwayLane_. The reason for not including the edge part of the abstraction is because we allow data-to-data property, data-to-object property, and object-to-object property abstractions. In previous example, the data property _:isMotorway_ is mapped to the object property _rdf:type._ Hence, we did not consider edges for Definition 4.

---

> > ### Comment · AnonReviewer1 · 2021-01-31
> > **An updated paper should be accepted, but not the original**
> >
> > Thank you for addressing my and the other reviewers' questions so thoroughly. Assuming these answers are incorporated into the text I will suggest accepting this paper for the conference.  I am not familiar with the details of the submission procedure, and if it is not possible to change the paper at this stage, I still suggest rejecting the original paper.

---

### Official Review · AnonReviewer4 · 2021-01-13
**The work is relevant to the research track. Main concerns are the lack of a thorough experimental study and the lack of formalization of some aspects of the work.**

**Confidence:** 4
**Impact:** 3
**Design And Technical Quality:** 3

**Review:**

The work addresses an interesting use case related to autonomous driving and the use of HD map data for issuing driver take-over requests. It applies an integrated system of ontologies and rules to a workflow for the semantic enrichment of map data, and the quality violation detection and violation handling of this data. The paper is in general well written but some specific parts are hard to follow (see the detailed comments). Also, it should include  formalization of the syntactic variant of the Datalog rules that are used. This formalization should also include the integration with the concepts in the ontologies.

The main concern is the lack of an experimental study. The study on performance of the workflow is done on the first two stages (it is not clear why the performance of the third stage was not evaluated), however its description is very limited, it is not clear how the 10 datasets were generated and the analysis of the performance results gives only an idea of the average time and size and this does not indicate the efficiency or scalability of the methods and techniques applied. The authors claim that they have evaluated the correctness of violation handling but it is more a proof of concept where only one of the possible violation handling cases is evaluated.

Some more detailed comments are the following:
- The RDF graph definition (Definition 1), and the RDF Isomormisphm definition (Definition 3) should include the corresponding references.
- Page 7, for  the template VRRT it is not clear what does "d" denote in hasReason(d,"r"),  in the example the predicate is hasReason(v,"MinCardinalityError").
- For topology checking rules there is an algorithm for full coverage checking. However, a declarative rule would be expected and it has not been defined.
- In the check full coverage algorithm it refers to the use of a FILTER in a VRRT for lines 4-7 and 10-13, this is not obvious, it should be clarified.
- it is not clear why a rule template is defined for violation detection but not for violation handling. For violation handling very specific examples are given but it is not clear if a generalization can be made for rules for violation handling.



**Anonymity:**

Yes, I would like my review to remain anonymous.

**Rating:**

-1: Weak Reject

**Reuse And Availability:**

2: Low

**Strong Points:**

1. The work addresses a current problem related to autonomous driving (AD) using HD map data and using ontologies and reasoning for violation detetection and handling.
2. The related work is clearly described and also the contribution with respect to this related work.
3. The work develops two new ontologies which may be relevant for this specific field: (a) HD Map ontology and (b) Map Quality Violation Ontology.
4. The work is an interesting use case for the integration of OWL ontologies and rules.
5. The work provides the application of  formalisms such as graph aggregation and graph decomposition.


**Subreviewer:**

I submitted this review.

**Weak Points:**

1. The work states that it has evaluated the corrrectness of the violation handling stage of the workflow. However, it shows a very limited evaluation that is more a proof of concept than a correctness evaluation (only evaluates the lane group ID uniqueness violation).
2. The description of evaluation of the performance of the semantic enrichment and violation detection stages is not well defined and the results are not analyzed in depth. It says that the map data is split into 10 data sets, however from Figure 9 one can infer that the datasets are of incremental size, what are the increments? As to the analysis of results, the average time is less relevant than the time increase in relation to the dataset size.
3. There is a lack of formalization of the Datalog rules that are used. It does mention that it is a syntactic variant developed in RDFox. This syntactic variant should be formalized since there is the use of functions in the rules.
4. Because of this lack of formalization there is not a clear definition of the integration of the ontology and rule systems. It is clear from the examples that classes and properties are predicates in the rules but this should be formalized.
5. The resources used in the experiental study are not available so their reuse is not possible.

---

> ### Author Rebuttal · Authors · 2021-01-29
>
> ## Response to AnonReviewer4
>
>
> Thank you for your helpful comments, which we will carefully address should the paper be accepted. We address raised questions below:
>
> &quot;_For topology checking rules there is an algorithm for full coverage checking. However, a declarative rule would be expected and it has not been defined.&quot;_
>
> We have included the corresponding rules in attached technical report that we refer to at the end of the introduction.
>
> &quot;_it is not clear why a rule template is defined for violation detection but not for violation handling. For violation handling very specific examples are given but it is not clear if a generalization can be made for rules for violation handling.&quot;_
>
> Violation handling seems to depend on more intricate context information that cannot easily be generalized. Note also that the presented approach is the first to show that some errors can be fixed based on context knowledge and reasoning.
>
> &quot;_The description of evaluation of the performance of the semantic enrichment and violation detection stages is not well defined and the results are not analyzed in depth. It says that the map data is split into 10 data sets, however from Figure 9 one can infer that the datasets are of incremental size, what are the increments? As to the analysis of results, the average time is less relevant than the time increase in relation to the dataset size.&quot;_
>
> We use tile-based real map data. Each tile has fixed length of borders. The data size of a tile depends on the density of roads and lanes contained in it. In Fig.9, we sort the data based on the size of each tile to see the impact of the data size on the computation time. As you pointed out, the computation time increases with respect to the data size. Regarding the average time, since the size range for highways is similar, the &quot;average&quot; of this range can give an approximation for the typical computation time for a tile along a highway.
>
> &quot;_There is a lack of formalization of the Datalog rules that are used. It does mention that it is a syntactic variant developed in RDFox. This syntactic variant should be formalized since there is the use of functions in the rules.&quot;
>
>  &quot;Because of this lack of formalization there is not a clear definition of the integration of the ontology and rule systems. It is clear from the examples that classes and properties are predicates in the rules but this should be formalized.&quot;_
>
> Indeed, we introduced the rules mainly by examples, but we could switch to a formal definition instead if that is considered important.
>
> &quot;_The resources used in the experiental study are not available so their reuse is not possible.&quot;_
>
> We are planning to make the ontology and rules available once the BMW internal review process is completed. The map data is, unfortunately, property of the map data provider (e.g. HERE), but we hope that other users of map data can already benefit from the ontology and rules.

---

### Official Review · AnonReviewer5 · 2021-01-14
**The paper describes a new ontology-based method to verify the quality of high-definition digital maps used for autonomous driving, and to suggest corrections for map errors.**

**Rating:** 1
**Confidence:** 4
**Impact:** 4
**Design And Technical Quality:** 5

**Review:**

The paper describes a method that uses OWL2 RL ontologies and Datalog rules to identify and fix violations in map data used in autonomous driving (AD) settings. The presented Map Quality Violation Ontology combined with constraint violation rules are used to detect violations in map data. The paper then presents strategies to rectify violations using graph aggregation and decomposition. The approach is evaluated in terms of performance of violation detection and correctness of violation handling. The results presented show that the approach can efficiently identify and fix two errors discovered in example map data.

The quality and clarity of the writing are great, it was a pleasure to read. The contribution of the paper is both original and impactful for the purpose of rectifying errors in road map data used in AD. It would have been good to test the approach in a larger corpus with more errors to handle. There are no links to the ontology, the rules, or the map data used in the evaluation, so reproducibility is questionable. Nonetheless, this is very interesting work worthy of presentation at ESWC.

Typos:
- Section 4.2: "via the hasResovledObject object property" -> "hasResolvedObject"
- Section 6: "knowledge-spatial architecture [25], And test the approach" -> "and"

**Anonymity:**

Yes, I would like my review to remain anonymous.

**Reuse And Availability:**

2: Low

**Strong Points:**

- Excellent writing and overall paper structure.
- Sound and compelling solution to ensure map data quality for autonomous driving purposes.

**Subreviewer:**

I submitted this review.

**Weak Points:**

- Small corpus used in the evaluation.
- The Map Violation Ontology, violation constraint rules, and the map data test corpus are not provided.
- Figure 8 is somewhat difficult to read. Spacing out elements in the diagram should make it more legible.
- The purpose or impact of testing the approach in Robot Operating System isn't clear.

---

> ### Author Rebuttal · Authors · 2021-01-29
>
> ## Response to AnonReviewer5
>
>
> Thank you for your helpful comments, which we will carefully address should the paper be accepted. We address raised questions below:
>
> &quot;_The Map Violation Ontology, violation constraint rules, and the map data test corpus are not provided.&quot;_
>
> We are planning to make the ontology and rules available once the BMW internal review process is completed. The map data is, unfortunately, property of the map data provider (e.g. HERE), but we hope that other users of map data can already benefit from the ontology and rules.
>
>
> &quot;_The purpose or impact of testing the approach in Robot Operating System isn&#39;t clear.&quot;_
>
> Autonomous driving systems are developed and simulated in the Robot Operating System (ROS) environment. Hence, porting the prototype into ROS is the next step towards a deployment in a real autonomous car. This requires the use of only C++ components, which is one reason for using RDFox as underlying reasoning system. Nevertheless, some further components and APIs still need to be adapted to be fully compatible with ROS.

---

### Official Review · AnonReviewer3 · 2021-01-17
**Review of article "Ontology-Based Map Data Quality Assurance"**

**Rating:** 1
**Confidence:** 3
**Impact:** 2
**Design And Technical Quality:** 3

**Review:**

The authors propose a semantic approach to enrich map metadata, and to automatically detect and correct map errors. The paper is well written and well organized, and the proposed approach is interesting and well described.

Pros:
- The authors clearly describe the domain, and define the problem.
- The technical description of the approach is clear.

Cons:
- The evaluation is perhaps a bit weak.
- It is not clear if the proposed approach has been deployed on a real autonomous vehicle or if it is only a prototype.

**Anonymity:**

Yes, I would like my review to remain anonymous.

**Reuse And Availability:**

2: Low

**Strong Points:**

- The authors clearly describe the domain, and define the problem.
- The technical description of the approach is clear.

**Subreviewer:**

I submitted this review.

**Weak Points:**

- General comment: is the method deployed in a real autonomous vehicle? If yes, please provide some description of the deployment; if no, please explain why.
- General comment on the proposed method (Figure 8 and 9): does the proposed method work offline (detecting and correcting maps errors before the maps are used), or does it work in real time while the vehicle is in autonomous driving mode? If it works in real time, are the performance described in the evaluation sufficient (~4 seconds seems a long time to me)?
- Some concerns over the evaluation:
  - General concern: How common are the errors that you consider in the evaluation in real world maps? In other words, how often can your method effectively detect and correct common real-world maps errors? Can you provide some quantitative information?
  - Section 5.2: "We consider the use case of lane group ID uniqueness violations": how significant is this use case? Can you provide some evidence or quantitative information that show that the use case is relevant and important. For example, if we consider the map of one sufficiently large country (say for example Germany): how often does this problem occur?
  - Table 3, row 2: is this dirty data after graph aggregation? if yes, then I understand why it is equal to row 1, but I suggest you specify "dirty data after graph aggregation". Instead, if this is the original dirty data, then I am confused and I do not understand why it is equal to row 1.
  - Section 5.2: "We applied graph aggregation over both the ground truth and dirty data": why do you apply graph aggregation over ground truth?
- Some minor language problems:
  - Section 3.1: "The road and lane model_s_ are fundamental..."
  - Section 4.3, page 11, just below figure 7: "... which causes l11 and l21 to _combine_ into ..."
  - Section 5.2: "We evaluated the resolution strategy over two lane groups allocated to different links and both _containing_ only one lane"

---

> ### Author Rebuttal · Authors · 2021-01-29
>
> ## Response to AnonReviewer3
>
>
> Thank you for your helpful comments, which we will carefully address should the paper be accepted. We address raised questions below:
>
> &quot;_General comment: is the method deployed in a real autonomous vehicle? If yes, please provide some description of the deployment; if no, please explain why.&quot;_
>
> The method is developed as part of a research project and not yet deployed in a real autonomous vehicle. Until the results of a research project can go into production, it may easily take a couple of years. At the current stage, we are working on deploying the prototype in an autonomous driving simulation environment, which requires a deployment in the Robot Operating System (ROS).
>
> &quot;_General comment on the proposed method (Figure 8 and 9): does the proposed method work offline (detecting and correcting maps errors before the maps are used), or does it work in real time while the vehicle is in autonomous driving mode? If it works in real time, are the performance described in the evaluation sufficient (~4 seconds seems a long time to me)?&quot;_
>
> The aim of the proposed method is to work both offline and in the car. HD maps are received continuously from map data providers (e.g. HERE) on a tile-basis and stored in the BMW back-end system. The back-end system will detect and correct map errors (offline) before sending to data the car. The car itself will also have learned maps whose quality needs to be check on-board (in the car). About the performance (4 seconds) considering in-car scenarios, we tackled this point by pre-fetching map tiles ahead of the route and by a parallel processing strategy. Since the checking happens a good deal before the car is actually using the data, they system can handle 4 second cost. We also assume that there is space for further improvement and one could also use a restricted set of rules in the car that only checks for the most critical errors.
>
> &quot;_General concern: How common are the errors that you consider in the evaluation in real world maps? In other words, how often can your method effectively detect and correct common real-world maps errors? Can you provide some quantitative information?&quot;_
>
> HD maps are received continuously (e.g., each 5 minutes) from map data providers and stored in the BMW back-end system. Whenever a tile is received, the system needs to detect errors and fix them in the back-end system. Errors occur quite frequently, so the issue if detecting and, ideally, fixing them automatically is very important. Note that our approach is the first that not only detects map errors, but can also fix some of the errors. Unfortunately, we have no authority to publish the exact quantitative information about map data errors in public.
>
> &quot;_Section 5.2: &quot;We consider the use case of lane group ID uniqueness violations&quot;: how significant is this use case? Can you provide some evidence or quantitative information that show that the use case is relevant and important. For example, if we consider the map of one sufficiently large country (say for example Germany): how often does this problem occur?&quot;_
>
> Lane group uniqueness violation is one of the top 3 occurring violations. About 5% of the tested tiles contain a lane group ID issue.
>
> &quot;_Table 3, row 2: is this dirty data after graph aggregation? if yes, then I understand why it is equal to row 1, but I suggest you specify &quot;dirty data after graph aggregation&quot;. Instead, if this is the original dirty data, then I am confused and I do not understand why it is equal to row 1.&quot;_
>
> Table 3 row 2 is indeed the dirty data. We will change the table title as you suggested to specify _&quot;dirty data after graph aggregation&quot;._
>
> &quot;_Section 5.2: &quot;We applied graph aggregation over both the ground truth and dirty data&quot;: why do you apply graph aggregation over ground truth?&quot;_
>
> The reason we apply graph aggregation over ground truth is that we use graph aggregation to aggregate several lanes into a continuous longer lane. When the data has a lane group ID duplication issue, the aggregated lane is the same as the aggregated result using ground truth. This proves that graph aggregation can tolerate errors in the data. One could, of course, argue whether ground truth (in the error-free scenario) is the data as originally provided or whether ground truth refers to the data after aggregation. We chose to consider the error-free originally provided data the ground truth.

---

> > ### Comment · AnonReviewer3 · 2021-02-09
> > **Answer to Rebuttal**
> >
> > Thank you for your clarification. I encourage you to add them to the final version of your paper if it is accepted.

---

### Decision · Program_Chairs · 2021-02-23

**Decision:**

Accept

**Comment:**

The majority of reviewers have recommended that this paper be accepted into the proceedings. This paper addresses an interesting topic, however, there is also a consensus on certain weaknesses of the paper. As such, while our final recommendation is _Accept_, we also strongly encourage the authors to incorporate the feedback of the reviewers into the final version of their paper. Additionally, make sure that there is a pointer to data artifacts (or where they will be), so that this work is discoverable and a (even more) useful resource in the future.